# The Effect of Autologous Platelet Rich Plasma on Endometrial Receptivity: A Narrative Review

**DOI:** 10.3390/medicina61010134

**Published:** 2025-01-15

**Authors:** Milan Stefanović, Predrag Vukomanović, Ranko Kutlesic, Milan Trenkić, Vanja Dimitrov, Aleksa Stefanović, Vladimir Cvetanović

**Affiliations:** 1Faculty of Medicine, University of Niš, Blvd. Dr Zoran Đinđić 81, 18000 Niš, Serbia; predragvukomanovic@yahoo.com (P.V.); trenkic@gmail.com (M.T.); vanjadimitrov@gmail.com (V.D.); lexstef97@gmail.com (A.S.); vladimircvetanovic1991@gmail.com (V.C.); 2Gynecology and Obstetrics Clinic, Clinical Center Niš, Blvd. Dr Zoran Đinđić 48, 18000 Niš, Serbia

**Keywords:** platelet-rich plasma, growth factors, endometrial thickness, endometrial receptivity, pregnancy rate

## Abstract

*Background and Objectives*: Autologous platelet-rich plasma (PRP) transfusions are a relatively new treatment method used in different fields of medicine, including the field of reproductive medicine. One of the applications of these concentrated platelet infusions is the treatment of endometrial receptivity, which is a key factor for embryo implantation. There are implications that PRP infusions can lead to increased endometrial thickness, endometrial receptivity, and significantly elevated clinical pregnancy rates. Our objective is to briefly understand what PRP is and to, through a narrative review, summarize the findings from studies focused on evaluating the benefits of PRP infusions to treat thin endometrium with the goal of achieving better endometrial receptivity. *Materials and Methods*: Reference data was searched using Medline, PubMed, and EMBASE to identify reports from 2015 to 2024. The combination of search words used was “PRP” and “platelet-rich plasma” with “thin endometrium”, “endometrial receptivity”, “endometrial thickness”, and “endometrial implantation”. Obtained articles were screened, and suited studies (randomized controlled trials, case reports, case series, pilot studies, and reviews) were included in the present review. Reports not available in the English language were eliminated from the current review. *Results*: The results from most of the reviewed studies showed a positive effect of autologous PRP infusions on increasing endometrial thickness, enhancing endometrial receptivity, and elevating clinical pregnancy rates. The majority of the evaluated findings revealed endometrial thickness > 7 mm (increased endometrial thickness was observed in each evaluated study) following the PRP treatment. More than 50% of the evaluated studies resulted in enhanced endometrial thickness, increased endometrial receptivity, and an elevated pregnancy rate after the PRP application. *Conclusions*: Autologous PRP infusions for treating endometrium are a relatively new method that has shown promising results. Its major strengths are availability and proper application, which eliminates possible immunological reactions or disease transmission. The main drawbacks are not enough data on safety (i.e., its effect on endometriosis) and the lack of uniformity in the PRP preparation, which would provide optimal standardized quality and quantity of the PRP product and, thus, optimal treatment results.

## 1. Introduction

The use of platelets in regenerative medicine has gained popularity in the past few decades to promote accelerated physiologic healing and restoration of function. Growth factors released from platelets have a role in the process of tissue regeneration at every stage, ensuring sufficient endogenous growth factors essential to tissue healing and regenerative processes [1]. As a main component of APCs (autologous platelet concentrates), platelets contain more than 1500 bioactive molecules, which are involved in cell proliferation, angiogenesis, and matrix remodeling [2]. Due to their easy preparation and their autologous nature, APCs are extensively used in different regenerative procedures [3,4].

Platelet-rich plasma (PRP), as a first generation of APCs [5], has demonstrated a key role in wound healing, together with cell proliferation and differentiation, because of the various growth factors released from activated platelets [2]. In recent years, PRP treatment gained focus on reproductive medicine, especially on infertility. Embryo quality and the receptivity of the endometrium represent the main factors that may determine positive implantation outcomes and positive rates of successful pregnancies [6]. Earlier studies [7,8] also showed that endometrial receptivity is a great factor for a majority of negative implantation outcomes, with endometrial thickness (7 mm being the lower limit for implantation) being a vital factor for endometrial receptivity [9]. Furthermore, different approaches have been conducted to improve endometrial receptivity, including the application of estrogen, vitamin E, peripheral blood mononuclear cells, or human chorionic gonadotropin [9,10,11], without significant success.

The application of granulocyte colony-stimulating factor and stem cells resulted in induced endometrial growth. However, the administration of these factors is limited due to uncontrolled differentiation and endometrial growth [12]. Compared to that, PRP application revealed useful effects by increasing the endometrial thickness and receptivity [13,14,15].

Recent findings suggest a promising role of PRP in enhancing endometrial cell differentiation, promoting vascular regeneration, and, most importantly, increasing endometrial thickness. Even though the routine application of PRP is still in the preliminary application stage, we hypothesize that the released growth factors from platelets may have a great impact on enhancing endometrial thickness. By increasing endometrial thickness, we can speculate about the improvement of endometrial receptivity and clinical pregnancy rates. The ability of released growth factors from PRP may present the ultimate treatment in women with thin endometrium, decreased endometrial receptivity, and repetitive negative birth outcomes. However, since the potential role of PRP in reproductive medicine is being extensively studied, the results and conclusions still lack consistency worldwide.

Therefore, the current review intends to determine the potential therapeutic effect of PRP on endometrial receptivity and possibly resolve and identify critical gaps for future research, especially in women with thin endometrium. Also, the present review aims to provide potential mechanisms involved in endometrial thickness enhancement, improved endometrial receptivity, and positive clinical pregnancy rates, as well as to outline the current reference data of possible beneficial effects of PRP on endometrium receptivity and to point out some drawbacks of PRP application therapy.

## 2. Materials and Methods

During the preparation of the current narrative review, reference data was searched using Medline, PubMed, and EMBASE to identify reports from 2015 to 2024. The combination of search words used was “PRP” and “platelet-rich plasma” with “thin endometrium”, “endometrial receptivity”, “endometrial thickness”, and “endometrial implantation”. Obtained articles were screened, and suited studies (randomized controlled trials, case reports, case series, pilot studies, and reviews) were included in the present review. Reports not available in the English language were eliminated from the current review.

## 3. Review Results

### 3.1. Platelet Rich Plasma

Even though various blood-derived products have been used more than 40 years ago to provoke the process of wound healing, PRP is usually described as a first generation of APCs. This type of APC was proposed for the first time in 1998 [16] and is, in essence, an autologous concentration of human platelets prepared from fresh whole blood.

PRP represents a small volume of human plasma that contains supraphysiological concentrations of platelets. While normal platelet count ranges from 15,000 to 300,000 cells/µL, PRP accommodates 3 to 5 times higher platelet concentration than the levels found in normal plasma [17]. Platelets are small and anucleate cells (without the possibility of reproducing), with a life span of around 7 to 10 days and stem from megakaryocytes in the bone marrow [5]. Since they have the smallest density compared to the other blood cells, centrifugation of the whole blood usually results in the separation of platelets at the top of an aggregate, which enables their easier extraction and subsequent clinical use.

The essential property of PRP is a very high concentration of platelets that, when activated, secrete a variety of different growth factors (6–8 times above the physiological concentrations). The high concentration of platelets and, thus, growth factors represent the key mechanism that stimulates the healing process. In line with this, it has been shown that depleted platelet amounts in PRP result in the reduced therapeutic effect of PRP [18]. Furthermore, besides growth factors, platelets can synthesize and secrete more than 1500 different bioactive molecules involved in the process of regeneration [2]. All of these molecules are able to activate different target cells (including mesenchymal stem cells) included in the regenerative process. These growth factors are stored within platelets in specific granules (alpha, delta, and lambda). Delta and lambda granules mainly contain biomolecules responsible for clotting and host defense. On the other hand, each platelet contains from 50 to 80% alpha granules, and they are the most abundant platelet granule type [5]. These granules enclose more than 300 diverse proteins [19] associated with cell growth and adhesion, as well as inflammation and clotting. However, the most prominent molecules secreted by alpha granules are different growth factors, including platelet-derived growth factor (PDGF), vascular endothelial growth factor (VEGF), transforming growth factor β (TGF-β), fibroblast growth factor (FGF), epidermal growth factor (EGF), connective tissue growth factor (CTGF), and insulin-like growth factor 1 (IGF-1).

All these growth factors coordinate chemotaxis, cellular differentiation, proliferation, and mitogenesis [20]. Secreted growth factors stimulate angiogenesis and increase blood supply, which further boosts the influx of nutrients and the required cells necessary in the healing process. Additionally, the ‘collaboration’ between released growth factors may also occur, resulting in the activation of various intracellular signaling pathways that finally intensify the regeneration process [21].

Since PRP represents an autologous product obtained from the patient’s blood, the application of this kind of platelet concentrate excludes any potential risk of disease transmission, immune reaction, or contamination if done properly. Taking into account this specific characteristic, the broad application of PRP resulted in the formation of more than 40 different PRP production protocols described to this day [5]. As of now, there is no uniform consensus or standardized protocol for PRP preparation from autologous whole blood, even though the application of PRP is firmly associated with its clinical efficacy [22].

Nevertheless, the preparation of autologous PRP includes two main steps: separation and concentration. Following the collection of blood into sterile tubes coated with an anticoagulant to prevent possible platelet activation. During the first step, blood is centrifuged for several minutes at room temperature, which leads to blood separation thanks to the various densities of blood cell components. This results in three distinct layers: the bottom layer, which contains red blood cells; the top layer, named platelet-poor plasma (PPP); and the middle layer, also called the buffy coat, which contains platelets and white blood cells [5,23]. The formed buffy coat and PPP are collected and transferred into another sterile tube for a second (concentration) centrifugation step, this time without any anticoagulants, in order to increase the platelet concentration. Usually, at the end of the second centrifugation step, platelets are located at the bottom of the centrifugation tube, allowing their easy extraction in plasma suspension (Figure 1).

Finally obtained PRP may be mixed with platelet activators (thrombin or calcium chloride) that induce platelet activation, the formation of platelet gel, and the secretion of different growth factors. Platelet activation may occur immediately after production or later since it has been shown that PRP persists viable for the next five days after production [24].

An important thing to mention is that platelet activation results in releasing the majority of growth factors within the first hour, of which around 95% are released in the first 10 min, indicating that the time of PRP is mainly responsible for PRP efficiency [2]. Endogenous activation of the PRP (platelet contact with collagen or cells’ basement membrane or thrombin) may increase the time that the target tissue is exposed to the growth factors. Additional clinical studies are needed to evaluate the PRP efficiency by endogenous compared to exogenous platelet activation.

On another note, due to the numerous and various protocols used for PRP production, the total amount of platelets in the final PRP product could vary, which, in turn, makes PRP efficiency comparison among diverse studies more complex. This may present one of the main shortcomings of PRP. In line with this, previous studies demonstrated different minimal amounts of platelets 800–1200 × 10^9^ cells/L [18], 1000 × 10^9^ cells/L [25], or 300 × 10^3^ µL [26] within PRP to be effective. Moreover, depending on which protocol has been used for PRP production, different amounts of other blood cells may be found in the final PRP.

Generally, an obtained PRP sample, besides platelets (95%), may contain erythrocytes (4%) and white blood cells (1%) [27]. Therefore, according to the presence or absence of white blood cells, two different PRP types have been proposed: leukocyte-rich PRP (L-PRP) and leukocyte-reduced PRP (P-PRP) [5]. The influence of white blood cells on the PRP efficiency is still unclear. Some studies proposed a positive effect by eradicating potentially present microorganisms, while other reports demonstrated the increased intensity of the inflammatory process and reduced regenerative PRP potential, as was reviewed earlier [2].

### 3.2. PRP—Its Effect on Endometrial Thickness and Endometrial Receptivity

Despite already being used in different fields, including maxillofacial surgery, dermatology, orthopedics, plastic surgery, sports, and veterinary medicine, in the last decade, autologous PRP transfusions have gained considerable attention in reproductive medicine. Several studies have reported that autologous PRP infusions can induce the proliferation of endometrial cells, enhance the morphology of the endometrium, improve angiogenesis, and reduce the secretion of inflammatory cytokines [15,28,29]. Therefore, the potential involvement of PRP in reproductive medicine is studied worldwide, especially the role of PRP in patients with a thin endometrium.

Similar to the other organs in the human body, the microenvironment of the endometrium (alterations of uterine glandular cells and modified activity of white blood cells) mainly regulates endometrial thickness as well as endometrial receptivity [30]. Additionally, several reports [31,32] have demonstrated that endometrial thickness of ≥7 mm usually results in higher pregnancy rates, while endometrial thickness < 7 mm is associated with decreased fertility rates [33,34]. Furthermore, thin endometrium (<7 mm) that has no response to hormonal therapy is also named ‘refractory endometrium’ and lacks optimal amounts of main angiogenic factors, especially VEGF [35]. Since substantial amounts of VEGF are stored within the concentrated platelets, PRP application may be a potential therapy for the refractory endometrium, compared to the standard treatment methods.

The first report to describe the use of a PRP method in patients with a thin endometrium was in 2015 [36] (Table 1). This study included five women who were on standard hormonal therapy (estradiol up to 12 mg/day), which resulted in poor endometrial response (endometrial thickness was <7 mm). Following the two intrauterine PRP autologous infusions (except one, who received a single infusion), endometrial thickness achieved 7 mm in all five women 48–72 h after autologous PRP infusions. Besides the increased endometrial thickness, all five women underwent embryo transfer and carried out successful pregnancies, except one, who had a missed abortion.

Another study showed that after double intrauterine autologous PRP infusion, with 48 h between the two doses, endometrial thickness increased ≥7 mm in 10 women with a thin endometrium, and in 5 out of 10 women pregnancy was confirmed after embryo transfer [37]. Results reported by Chang et al. [36] were later observed in similar studies, indicating an increase in endometrial thickness after PRP application [40,45].

Based on the obtained search results, the first randomized controlled study was conducted in 2018 [39]. This report included 83 patients with thin endometrium who were divided into two groups: a control group on the standard (estradiol) hormonal therapy (*n* = 40) and a PRP group (*n* = 43). The study results demonstrated that a single intrauterine autologous PRP infusion markedly increased endometrial thickness and subsequent implantation in more than 2/3 of the group (*n* = 23), and 10 women needed two autologous PRP infusions. However, even though the pregnancy rates were higher in women who received autologous PRP infusion compared to the control group, it was not statistically significant [39]. We would also like to highlight that 10 women from the control group and 7 women from the PRP group had a persistent thin endometrium that did not respond to either therapy.

These findings were confirmed by a later randomized controlled study [41], which involved 60 women with thin endometrium. Double intrauterine autologous PRP infusion showed a significant increase in endometrial thickness (>7 mm) and an increased the clinical pregnancy rate compared to the sham-catheter group.

Another randomized controlled study enrolled 120 women with thin endometrium and divided them into two groups: one with standard (estradiol) hormonal therapy and one with intrauterine PRP autologous infusions [46]. The study results demonstrated that intrauterine PRP autologous infusions provided significantly increased endometrial thickness (>7 mm) after 3 cycles, improved endometrial vascularization, and increased pregnancy rate after PRP application, compared to the group with standard hormonal therapy.

Taking into account their earlier results, Chang et al. [47] performed a prospective cohort study, which included 64 women with a thin endometrium (<7 mm). All the patients received standard (estradiol) hormonal therapy up to 12 mg of estradiol daily. If the endometrial thickness remained <7 mm, women would either receive intrauterine autologous PRP infusion (PRP group) or continue without PRP application (control group) to undergo embryo transfer, regardless of the thickness of the endometrium. The results of this study showed a significant increase in endometrial thickness, implantation rates, and increased clinical pregnancy rates in the PRP group compared to the control group. This study also measured the concentration of certain bioactive molecules and found that the PRP solution contains significantly higher values of PDGF-AB, PDGF-BB, and TGF-β compared to the peripheral blood [47].

Recently, a double-blind, randomized controlled study that included 97 women with thin endometrium was conducted, in which 49 women received autologous PRP infusion (PRP group) while 48 women received standard treatment (control group) without PRP application [48]. All participants received estradiol therapy (up to 8 mg/daily) until the endometrial thickness reached at least 7 mm. Regardless of the endometrial thickness achieved 7 mm or more, all women underwent embryo transfer. Intrauterine autologous PRP infusion was implemented 48 h before the embryo transfer. The findings of this study revealed that the chemical and clinical pregnancy rates were significantly higher in the PRP group than in the control group.

Furthermore, in another study [42], which involved 21 women with a thin endometrium (<5 mm) and standard hormonal (estradiol) therapy (up to 8 mg/daily), three intrauterine autologous PRP infusions were applied before the embryo transfer, regardless of the endometrium thickness. The obtained results showed that the majority (54% or 13 cases) of clinical pregnancies resulted in live births, and only 3 miscarriages were detected, indicating that PRP can enhance endometrial receptivity, regardless of the endometrial thickness [42].

A retrospective report with 85 women participating showed that intrauterine infusion of autologous PRP women (with endometrium < 7 mm) resulted in increased endometrial thickness and a significant increase in live birth rate in women with PRP treatment compared to the control group [49]. It should be mentioned that the current study had a small sample size and included a control group of previous embryo transfers, which may have contributed to the results.

A relatively new approach to improving endometrial thickness using PRP gel instead of intrauterine PRP infusion has been shown in different reports [44,47]. Using PRP gel has a couple of advantages over the standard intrauterine autologous PRP infusion. Namely, due to the specific anatomical uterine cavity position, liquid PRP is quickly eliminated following the infusion compared to the PRP gel, which may stay for a longer time period and diminish the loss of growth factors and platelets [50]. Additionally, in activated PRP, prominent growth factors (VEGF, PDGF, and TGF-β) are more concentrated than in non-activated PRP and may show continuous release for up to 7 days [5]. Angiogenesis intensity was shown to be more efficient when PRP gel was introduced compared to the standard intrauterine PRP infusion [51]. These findings were confirmed in a recent study, demonstrating that PRP gel application resulted in a significantly increased clinical pregnancy rate compared to the control group [44].

### 3.3. Certain Mechanisms of PRP Involved in Modulating Endometrial Thickness and Receptivity

Having in mind the previous findings, it seems that intrauterine autologous PRP infusions may represent a possible alternative for improving endometrial thickness, implantation, and chemical and clinical pregnancy rates. Nevertheless, larger randomized trials are necessary to clarify the potential role of PRP in future clinical practice. Different studies have been conducted to evaluate the possible PRP effect on endometrial regenerative capacity, which leads to positive endometrial receptivity and increased implantation rates. The key mechanism of action for PRP is based on the secretion of diverse growth factors following platelet activation. The most prominent growth factors from PRP (VEGF, PDGF, and TGF-β) are massively secreted after platelet activation, and they are able to induce tissue regeneration (which includes the endometrium) and stimulate the proliferation of endothelial cells [51]. In line with this, various proliferation markers in endometrial stromal cells (cytokeratin, ki67, and Hoxa10) have shown increased expression after PRP application [29]. Furthermore, Hoxa10 and leukemia inhibitory factor, which are necessary for endometrial development, increase their expression after PRP application [52]. Also, it has been shown that PRP induces the proliferation of mesenchymal stem cells and fibroblasts in the endometrium, which leads to increased endometrium thickness [53]. Released PDGF from PRP stimulates the remodeling of the extracellular matrix and induces cell migration to the endometrium [54]. These findings are confirmed by an in vitro study [53], where different types of cultured human endometrial cells, after the application of activated PRP, resulted in an increased migration rate of all cells involved in the regeneration process.

Released VEGF from activated autologous PRP induces neoangiogenesis, subsequently increasing blood supply for endometrial growth and nutrition supply and providing excellent conditions for future implantation [55]. Another report demonstrated that PRP infusion may enhance endometrial blood supply and provide better endometrial receptivity [54]. An in vitro study revealed that mesenchymal stem cell cultivation with activated PRP may promote higher expression of endoglin (CD105) and CD146 molecules, which are included in the process of angiogenesis [56]. Different animal studies reported that PRP application increased the expression of proangiogenic factors, including hypoxia-inducible factor 1-alpha (HIF1α), hypoxia-inducible factor 2-alpha (HIF2α), IGF-1, and hepatocyte growth factor (HGF) [57,58].

Following the autologous PRP application, the levels of mRNA of pro-inflammatory cytokines (IL-8, IL-6, IL-1β) were significantly decreased [56], together with inhibited fibrosis [15]. Inhibiting fibrosis and tissue damage by controlling the intensity of inflammatory response may provide the reparation and regeneration of the endometrium. Furthermore, the application of autologous PRP significantly reduces the number of NK cells, CD8 cells, and Th1 cells, which may also impact the intensity of the inflammatory process [59]. On the other hand, the elevated expression of some matrix metalloproteinases (MMP1, MMP3, and MMP7) after treatment with autologous PRP may induce tissue reparation by remodeling the extracellular matrix [53]. Even though the vascularization of the endometrium has been shown to be one of the key factors that determine the implantation capability of the human endometrium, further studies should be conducted to determine the efficiency of PRP in different patients.

The efficacy of autologous PRP treatment may differ depending on the way the PRP therapy was applied. Namely, up to now, there are two different ways for intrauterine administration of autologous PRP, including the application to the endometrial surface (intrauterine perfusion) or sub-endometrial administration. Due to the anatomical characteristics of the uterine cavity, application to the endometrial surface may result in a premature loss of autologous PRP compared to the subendometrial PRP administration. The extended presence of subendometrial application of autologous PRP provides a higher and longer release of growth factors necessary for endometrial reparation and regeneration. However, there is still no consensus about which way of PRP application is more efficient, which is confirmed in a recent study showing that there is no significant difference in embryo transfer outcomes by using either one of these two different methods for PRP application [60]. Even though treatment with autologous PRP has beneficial effects on endometrial receptivity, more studies are necessary to precisely determine the efficacy of this method.

## 4. Discussion

The current review brings an overview of different study designs, ranging from case reports to case randomized control studies, of the autologous PRP treatment in women with reproductive potential. Since there are a limited number of randomized controlled studies and systematic reviews about the present subject, we selected a narrative review for the evaluation of the effect of the autologous PRP treatment of endometrial receptivity. Also, we are fully aware that there is no standard method for data separation in narrative review; thus, the possibility of prejudice cannot be excluded.

Furthermore, the plethora of diverse methods for autologous PRP preparation and inconsistency in PRP kits may lead to different concentrations of platelets, leukocytes, and, finally, growth factors in the sample, which is one of the main PRP shortcomings in adequately interpreting the data and results (Table 2).

However, it should be pointed out that a recent report [61] demonstrated no significant changes in increasing endometrial thickness following the PRP application compared to the control group with a conventional therapeutic approach. Moreover, another study reported no significant difference after PRP application in clinical and biochemical pregnancy rates in women with repeated implantation failure (RIF) [62]. Even though there is growing evidence about the positive effect of PRP on endometrial thickness and endometrial receptivity, insufficient clinical studies have proposed a positive PRP effect on RIF. One of the reasons for such inconsistent results may be that most of the PRP studies are retrospective reports with small sample sizes and high heterogeneity. Also, intrauterine PRP application may result in local endometrial injury, with the necessary appropriate control group (sham catheter group) that lacks most of the studies. In line with this, randomized control trials with larger sample sizes are needed to provide efficiency of PRP application in future.

During a frozen embryo transfer (FET) cycle, intrauterine PRP infusions require to be repeated two, three, or more times before endometrial thickness reaches ≥7 mm, confirmed by multiple studies [40,49,63,64]. Furthermore, a recent study has shown that only 18 women out of 51 have not reached an endometrial thickness of 7 mm [65]. The cryopreservation process of PRP may cause the loss of some growth and clotting factors, together with loss of clot elasticity, using freeze-thawed PRP [66]. Since there is only one retrospective study using freeze-thawed PRP for enhancing endometrial thickness [44], obtained results should be interpreted with caution since it is a retrospective study, and the possibility of bias cannot be excluded.

Therefore, the uniformity in preparation methods for PRP, high-quality studies with larger sample sizes, and further research of the precise mechanism of PRP treatment are basic imperative to evaluate the benefits and potential risks associated with PRP application.

Even though prospective studies exist, the total number of participants is relatively small, with a short observation time. Further studies are necessary to evaluate the safety of the offspring with a larger sample size. Also, it should point out the possible impact of autologous PRP treatment in women with ectopic endometriosis, where PRP administration may provoke uncontrolled endometrial proliferation and the risks associated with it. Therefore, larger randomized control studies with larger sample sizes must determine the precise mechanisms involved in autologous PRP treatments. It will only be possible to eliminate the potential risks linked with autologous PRP treatments.

## 5. Conclusions

The intrauterine infusion of autologous PRP represents a relatively new approach in reproductive medicine, especially with the intent of increasing endometrial thickness and endometrial receptivity. This therapeutic approach is a type of low-cost therapy that can be easily prepared and administered in a short-time period. Since it is prepared from the patient’s blood, it eliminates possible immunological reactions or disease transmission.

PRP contains huge amounts of growth factors and cytokines needed for tissue healing and regeneration, which makes it great potential for different indications in reproductive medicine. However, there is a lack of uniformity in the PRP preparation, which provides differences in the quality and quantity of the PRP product. In addition, there is no standard procedure for timing and no standardized dosage that is considered optimal for using autologous PRP to achieve the best results.

Having all of this in mind (the lack of uniformity in preparation and dosage/timing standardization) and the scarcity of high-quality follow-up studies make it difficult to obtain consistent results.

Moreover, limited, mainly retrospective studies, especially in patients with RIF and FET, do not provide enough data for proper conclusions, and the possibility of bias cannot be excluded.

Further research about the precise mechanism of PRP therapy may provide a better understanding of the role of some PRP bioactive components, which may, in return, lead to conclusive evidence and prevent potential risks associated with PRP therapy.

Despite the demonstrated potential of autologous PRP in reproductive medicine, a complete understanding of the mechanisms by which PRP induces positive outcomes in reproductive medicine is necessary. Therefore, further prospective and randomized controlled studies are required to better understand the fundamental function of PRP and to establish the future guidelines for its use before usage can be recommended routinely.

## Figures and Tables

**Figure 1 medicina-61-00134-f001:**
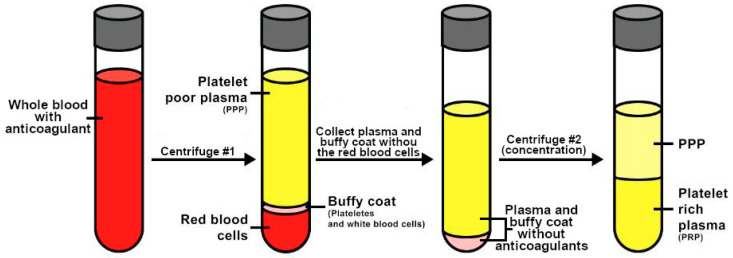
Simplified scheme of PRP preparation.

**Table 1 medicina-61-00134-t001:** Summary of studies evaluating the PRP infusions and its outcomes.

Author	Year	Endometrium Thickness After the Treatment	Main Results After the PRP Treatment
Chang et al. [36]	2015	>7 mm	Increased endometrial thickness and pregnancy rate
Zadehmodarres et al. [37]	2017	>7 mm	Increased endometrial thickness
Tandulwadkar et. [38]	2017	>7 mm	Increased endometrial thickness, receptivity, vascularity, and clinical pregnancy rate
Eftekhar et al. [39]	2018	>8 mm	Increased endometrial thickness, receptivity, and pregnancy rate
Kim et al. [40]	2019	4–9 mm	Increased endometrial receptivity, pregnancy, and birth rate
Nazari et al. [41]	2019	>7 mm	Increased endometrial thickness and chemical pregnancies
Frantz et al. [42]	2020	<5 mm	Increased clinical pregnancies
Agarwal et al. [43]	2020	>7 mm	Increased endometrial thickness, receptivity, and pregnancy rate
Wang et al. [44]	2024	6.1–7.2 mm	Increased endometrial thickness, receptivity, and pregnancy rate

**Table 2 medicina-61-00134-t002:** The main advantages and disadvantages of autologous PRP.

Advantages	Disadvantages
Inactivated PRP represents a completely autologous product	Diversity of PRP kits and different production procedures
Easy to prepare	No consensus about PRP optimal dose and timing for application
Possesses huge amounts of growth factors and cytokines needed for the regeneration process	Lacking the uniformity of the total amount of platelets and white blood cells
Spontaneous PRP activation reduces any possibility of immunological or hematology disorders	Application of anticoagulant may alter fibrin clot formation
Requires minimal costs	Possible reaction to bovine thrombin may result in bleeding disorders

## Data Availability

All the data used in the study are listed as references and are published articles. For any inquiries, please contact the corresponding author.

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
