# Peer review of "The Effect of Autologous Platelet Rich Plasma on Endometrial Receptivity: A Narrative Review"

_medicina, 2025, doi:10.3390/medicina61010134_

Round 1

Reviewer 1 Report

Comments and Suggestions for Authors

The study "The effect of autologous platelet-rich plasma on endometrial receptivity: A narrative review" presents some advantages and disadvantages of using autologous PRP infusions to increase endometrial thickness and receptivity. Yet, some specific comments and suggestions are provided below that need to be addressed.

Abstract:

1. Increase emphasis on the results section by providing a more detailed and comprehensive explanation.

Introduction:

1. The introduction is insufficiently detailed. This section lacks a clear rationale outlining the study's objectives and hypotheses. To address this, you need to expand upon the main aims of the study and provide a concise background in the introduction section. This will offer a clearer context for readers regarding the study's goals and the underlying hypotheses.

2. Please highlight the novel aspects of your research in this part.

Discussion:

1. The discussion section is too brief and lacks a balanced view. It does not adequately address conflicting evidence, particularly the perspective of researchers who argue that autologous PRP treatment may not offer significant benefits. A critical discussion incorporating both supportive and opposing viewpoints is essential to provide a comprehensive understanding of the topic.

2. Table 2: The listed advantages include apparent redundancy, as advantages 2 and 5 seem to be the same. I recommend reviewing these points and revising them for clarity to eliminate duplication and ensure a concise presentation.

Conclusion:

1. Recent systematic and narrative reviews published in 2024 have also explored the effects of autologous platelet-rich plasma (PRP) on endometrial receptivity. What differentiates this review from the existing literature? What novel insights or unique contributions does it offer? Currently, the introduction, discussion, and conclusion sections do not highlight the study's specific novelty or distinguishing features. Emphasizing these aspects would enhance the quality of the manuscript.

2. It would be beneficial to include a more practical outline for future studies in this field based on your results to make the conclusion more forward-looking.

Author Response

Dear reviewers,

Thank you very much for your valuable comments about our manuscript entitled: "The effect of autologous platelet-rich plasma on endometrial receptivity: A narrative review".

The answer for the reviewers is following.

Answer to the reviewer 1. Thank you very much for your constructive comments about our manuscript.

#1 "Abstract: 1. Increase emphasis on the results section by providing a more detailed and comprehensive explanation".

We fully accept your comment and we added in the Introduction section:" Majority of the evaluated findings revealed endometrial thickness >7 mm (even increased endometrial thickness was observed in each evaluated study), following the PRP treatment. More than 50% of evaluated studies resulted with enhanced endometrial thickness, increased endometrial receptivity together with elevated pregnancy rate, after PRP application".

#2 Introduction: 1. The introduction is insufficiently detailed. This section lacks a clear rationale outlining the study's objectives and hypotheses. To address this, you need to expand upon the main aims of the study and provide a concise background in the introduction section. This will offer a clearer context for readers regarding the study's goals and the underlying hypotheses.

We followed your comment and we changed the background by adding the following text:" Growth factors, released from platelets, supervise the process of tissue regeneration at every stage and ensuring sufficient endogenous growth factors essential to tissue healing and regenerative procedures [1]. Platelets, as a main component of APCs (autologous platelet concentrates), contain more than 1500 bioactive molecules, which are involved in cell proliferation, angiogenesis and matrix remodeling [2]. Due to their easy preparation and their autologous nature, APCs are extensively used in different regenerative procedures [3, 4].

Platelet rich plasma (PRP), as a first generation of APC [5], has demonstrated key role in wound healing together with cell proliferation and differentiation, which are provided by various growth factors released from activated platelets [2]. In recent years, PRP effectiveness switched focus on reproductive medicine, especially on infertility. Embryo quality and receptivity of the endometrium represent main factors which may determine positive implantation outcome and positive rates of successful pregnancies [6]. Earlier studies [7, 8] also showed that endometrial receptivity is a great factor for a majority of negative implantation outcomes, with endometrial thickness (7mm being the lower limit for implantation) being a vital factor for endometrial receptivity [9]. Furthermore, different approaches have been conducted to improve endometrial receptivity, including application of estrogen, vitamin E, peripheral blood mononuclear cells or human chorionic gonadotropin [9, 10, 11], without significant success. Also, application of granulocyte colony-stimulating factor and stem cells resulted with an induced endometrial growth. However, the administration of these factors is limited due to uncontrolled differentiation and endometrial growth [12]. On the other hand, PRP application revealed useful effect by increasing the endometrial thickness and receptivity [13, 14, 15].

 ….. Even the routine application of PRP is still in preliminary application stage, we hypothesize that released growth factors from platelets may have great impact in enhancing the endometrial thickness. By increasing the endometrial thickness, we can speculate about the improvement of endometrial receptivity and clinical pregnancy rates. The ability of released growth factors from PRP may represent ultimate goal in women with thin endometrium, decreased endometrial receptivity and repetitive negative birth outcomes. However, since potential role of PRP in reproductive medicine is extensively studied, conflicting results still lack consistency worldwide. Therefore, the current review intends to determinate the potential therapeutic effect of PRP on endometrial receptivity and possible resolves and identifies critical gaps for future research especially in women with thin endometrium. Also, the present review aims to provide potential mechanisms involved in endometrial thickness enhancing, improved endometrial receptivity, positive clinical pregnancy rates, as well as to outline the current reference data of possible beneficial effects of PRP on endometrium receptivity and to point out some drawbacks of PRP application therapy.

  1. Please highlight the novel aspects of your research in this part.

All the changes made are underlined.

#3 Discussion: 1. The discussion section is too brief and lacks a balanced view. It does not adequately address conflicting evidence, particularly the perspective of researchers who argue that autologous PRP treatment may not offer significant benefits. A critical discussion incorporating both supportive and opposing viewpoints is essential to provide a comprehensive understanding of the topic.

According to your suggestion we added new reference data and discussed about reported findings.

However, it should be pointed out that recent report [61] demonstrated no significant changes in increasing endometrial thickness, following the PRP application, compared to the control group with conventional therapeutic approach. Moreover, another study reported no significant difference, after PRP application, in clinical and biochemical pregnancy rates in women with repeated implantation failure (RIF) [62]. Even there are growing evidence about positive effect of PRP on endometrial thickness and endometrial receptivity, insufficient clinical studies proposed positive PRP effect on RIF. One of the reason for such inconsistent results may be because most of the PRP studies are retrospective reports, with small sample size and high heterogeneity. Also, intrauterine PRP application may result with local endometrial injury, with necessary appropriate control group (sham catheter group) which lacks in the most of the studies. In line with this, randomized control trials with larger sample size are needed to provide efficiency of PRP application in future.

Earlier studies confirmed that intrauterine PRP infusion requires to be repeated two, three or more times, before endometrial thickness reaches ≥7 mm [63, 64] during frozen embryo transfer (FET) cycle, which is also confirmed by other studies [65, 66]. Furthermore, recent study showed that only 18 women out of 51 have not reached endometrial thickness of 7 mm [67]. Cryopreservation process of PRP may provide the loss of some growth and clotting factors, together with loos of clot elasticity by using freeze-thawed PRP [68]. Since there is only one retrospective study by using freeze-thawed PRP for enhancing the endometrial thickness [69], obtain results should be considered with caution, since it is retrospective study and possibility of bias cannot be excluded. Therefore, diversity of PRP preparation procedure, lack of uniformity in preparation method and component content, high quality studies with larger sample size and further research of precise mechanism of PRP treatment, represents basic imperative to evaluate the potential risks with PRP application.

  1. Table 2: The listed advantages include apparent redundancy, as advantages 2 and 5 seem to be the same. I recommend reviewing these points and revising them for clarity to eliminate duplication and ensure a concise presentation.

We corrected the Table 2, according to your comments.

#4 Conclusion: 1. Recent systematic and narrative reviews published in 2024 have also explored the effects of autologous platelet-rich plasma (PRP) on endometrial receptivity. What differentiates this review from the existing literature? What novel insights or unique contributions does it offer? Currently, the introduction, discussion, and conclusion sections do not highlight the study's specific novelty or distinguishing features. Emphasizing these aspects would enhance the quality of the manuscript.

We followed your comments and added the following text in the manuscript:" Also, some conflicting results about PRP effectiveness may result due to inconsistency of PRP preparation, different component content as well as lack of high quality, follow-up studies with larger sample sizes. Moreover, limited, mainly retrospectives studies, especially in patients with RIF and FET do not provide enough data for proper conclusions and possibility of bias cannot be excluded. Further research about precise mechanism of PRP therapy may provide better understanding the role of some PRP bioactive components which may, in return, lead to conclusive evidence and prevent potential risks associated with PRP therapy".

Also this manuscript provides basic information about PRP preparation, potential advantages and drawback of the PRP therapy, especially focused to the endometrial thickness and receptivity. The article will likely attract interest from a broad audience, including clinicians and researchers in reproductive medicine, due to its focus on a low-cost, non-invasive therapeutic alternative. The potential clinical implications of PRP therapy are significant, making the work relevant and timely. Furthermore, as was stated by the Reviewer 2, "While the narrative review does not resolve a long-standing issue, it offers valuable insights and identifies critical gaps for future research. Overall, the article provides a useful synthesis of the current knowledge on PRP in reproductive medicine". 

We hope to hearing from You soon.

Best regards.

Reviewer 2 Report

Comments and Suggestions for Authors

The article explores the use of autologous platelet-rich plasma (PRP) in improving endometrial receptivity, a novel and emerging area in reproductive medicine. It addresses a well-defined question about PRP's potential to enhance endometrial thickness, receptivity, and pregnancy outcomes. This topic is of high relevance, offering significant advancements in knowledge for addressing thin endometrium and associated fertility challenges.

The work fits well within the scope of the journal, targeting readers interested in innovative therapeutic approaches in gynecology and reproductive health. It presents significant findings from a range of studies, highlighting the benefits of PRP in reproductive medicine. While the results are promising, the narrative review format introduces some limitations, as it lacks a systematic methodology for evaluating the quality and rigor of the included studies. Nonetheless, the hypotheses regarding PRP’s mechanisms, such as the secretion of growth factors like VEGF and PDGF, are well-supported by existing literature.

The article is well-organized and written in clear, concise language. It effectively describes PRP preparation techniques and presents findings from various study designs, ranging from case reports to randomized controlled trials. However, the variability in PRP preparation protocols and the absence of standardized methods are notable shortcomings. These factors complicate the interpretation of results and highlight the need for further research to establish uniform preparation guidelines.

The study design and technical aspects are sound, with sufficient details provided to allow replication of the PRP preparation and application methods. However, the lack of raw data or statistical analysis limits the robustness of the conclusions. The authors appropriately identify key limitations, including insufficient data on PRP safety and its long-term effects, particularly in conditions like endometriosis.

The article will likely attract interest from a broad audience, including clinicians and researchers in reproductive medicine, due to its focus on a low-cost, non-invasive therapeutic alternative. The potential clinical implications of PRP therapy are significant, making the work relevant and timely. While the narrative review does not resolve a long-standing issue, it offers valuable insights and identifies critical gaps for future research.

Overall, the article provides a useful synthesis of the current knowledge on PRP in reproductive medicine. However, its impact would be enhanced by addressing the variability in PRP preparation and conducting larger, controlled studies to strengthen the evidence base. The work is suitable for publication with minor revisions to improve the critical evaluation of included studies and to address methodological variability.

Author Response

It is very kind to hear such a nice word from you about manuscript we have submitted. Thank you for your valuable comments which helped us to improve the quality of the manuscript.

 #1 The work is suitable for publication with minor revisions to improve the critical evaluation of included studies and to address methodological variability

We accepted your comments and we added new text in our manuscript regarding the evaluation of some obtained results and methodological variability.

 " However, it should be pointed out that recent report [61] demonstrated no significant changes in increasing endometrial thickness, following the PRP application, compared to the control group with conventional therapeutic approach. Moreover, another study reported no significant difference, after PRP application, in clinical and biochemical pregnancy rates in women with repeated implantation failure (RIF) [62]. Even there are growing evidence about positive effect of PRP on endometrial thickness and endometrial receptivity, insufficient clinical studies proposed positive PRP effect on RIF. One of the reason for such inconsistent results may be because most of the PRP studies are retrospective reports, with small sample size and high heterogeneity. Also, intrauterine PRP application may result with local endometrial injury, with necessary appropriate control group (sham catheter group) which lacks in the most of the studies. In line with this, randomized control trials with larger sample size are needed to provide efficiency of PRP application in future.

Earlier studies confirmed that intrauterine PRP infusion requires to be repeated two, three or more times, before endometrial thickness reaches ≥7 mm [63, 64] during frozen embryo transfer (FET) cycle, which is also confirmed by other studies [65, 66]. Furthermore, recent study showed that only 18 women out of 51 have not reached endometrial thickness of 7 mm [67]. Cryopreservation process of PRP may provide the loss of some growth and clotting factors, together with loos of clot elasticity by using freeze-thawed PRP [68]. Since there is only one retrospective study by using freeze-thawed PRP for enhancing the endometrial thickness [69], obtain results should be considered with caution, since it is retrospective study and possibility of bias cannot be excluded. Therefore, diversity of PRP preparation procedure, lack of uniformity in preparation method and component content, high quality studies with larger sample size and further research of precise mechanism of PRP treatment, represents basic imperative to evaluate the potential risks with PRP application.

…. Also, some conflicting results about PRP effectiveness may result due to inconsistency of PRP preparation, different component content as well as lack of high quality, follow-up studies with larger sample sizes. Moreover, limited, mainly retrospectives studies, especially in patients with RIF and FET do not provide enough data for proper conclusions and possibility of bias cannot be excluded. Further research about precise mechanism of PRP therapy may provide better understanding the role of some PRP bioactive components which may, in return, lead to conclusive evidence and prevent potential risks associated with PRP therapy.

We hope to hear from You soon.

Best regards.